# Heterogeneous Catalysis of Ozone Using Iron–Manganese Silicate for Degradation of Acrylic Acid

**DOI:** 10.3390/molecules27154973

**Published:** 2022-08-05

**Authors:** Yue Liu, Congmin Wang, Rong Guo, Juexiu Li, Quan Zhao, Weiqiang Wang, Fei Qi, Haifang Liu, Yang Li, Huifan Zheng

**Affiliations:** 1School of Energy & Environment Engineering, Zhongyuan University of Technology, Zhengzhou 451191, China; 2State Key Laboratory of Urban Water Resource and Environment, School of Environment, Harbin Institute of Technology, Harbin 150090, China; 3Beijing Key Lab for Source Control Technology of Water Pollution, College of Environmental Science and Engineering, Beijing Forestry University, Beijing 100083, China

**Keywords:** catalytic ozonation, ozonation alone, iron–manganese silicate, hydroxyl radical, acrylic acid

## Abstract

Iron–manganese silicate (IMS) was synthesized by chemical coprecipitation and used as a catalyst for ozonating acrylic acid (AA) in semicontinuous flow mode. The Fe-O-Mn bond, Fe-Si, and Mn-Si binary oxide were formed in IMS on the basis of the results of XRD, FTIR, and XPS analysis. The removal efficiency of AA was highest in the IMS catalytic ozonation processes (98.9% in 15 min) compared with ozonation alone (62.7%), iron silicate (IS) catalytic ozonation (95.6%), and manganese silicate catalytic ozonation (94.8%). Meanwhile, the removal efficiencies of total organic carbon (TOC) were also improved in the IMS catalytic ozonation processes. The IMS showed high stability and ozone utilization. Additionally, H_2_O_2_ was formed in the process of IMS catalytic ozonation. Electron paramagnetic resonance (EPR) analysis and radical scavenger experiments confirmed that hydroxyl radicals (•OH) were the dominant oxidants. Cl^−^, HCO_3_^−^, PO_4_^3−^, Ca^2+^, and Mg^2+^ in aqueous solution could adversely affect AA degradation. In the IMS catalytic ozonation of AA, the surface hydroxyl groups and Lewis acid sites played an important role.

## 1. Introduction

Acrylic acid (C_3_H_4_O_2_, AA) composed of a vinyl group and a carboxyl group is a very fast polymerization of ethylene monomer that can perform homopolymerization and copolymerization. AA can be used in the production of synthetic resin, synthetic fiber, building materials, coatings, printing, and so on [1]. As a result, it is frequently detected in surface water and wastewater [2]. The presence of AA in water, even at low concentrations, can pose potential hazards to human health and ecology, since it is highly acidic in aqueous solution and difficult to biodegrade [3]. Therefore, effective methods for removing AA from water are highly recommended.

An advanced oxidation technology using a solid catalyst to accelerate ozone decomposition at room temperature to produce highly active intermediates such as •OH is known as heterogeneous catalytic ozonation [4]. In this technology, the catalyst can be added to the reactor at one time, and is easily separated from water and recycled [5]. Since this technology is easy to apply in practical water treatment projects, it has become a hot research topic in the field [6]. Metal oxides [7,8], metal silicates [9], metal or metal oxide supported on the carrier [10,11,12], and natural minerals with mesoporous [13,14] characteristics have been studied extensively in the past few years as heterogeneous ozonation catalysts because they can generate reactive oxygen species (ROS) and degrade refractory organic pollutants by catalyzing ozone decomposition. Among these catalysts, metal silicates have received great attention as heterogeneous catalytic ozonation catalysts because they have ordered mesoporous channels, large specific surface area, good chemical stability, a low cost, and abundant hydroxyl groups on the surface [15]. Research has demonstrated that surface hydroxyl and surface-active species of metal silicates can accelerate ozone decomposition in water, producing strong oxidizing •OH that increases organic pollutant degradation [15,16,17,18]. According to reports, adding different metal ions to the silicate preparation process affects not only the crystal structure of the silicate but also the electron transport capacity of catalyst surfaces and interfaces and the yield of free radicals during catalytic ozonation [19].

With this in mind, in this study we synthesized an iron–manganese silicate catalyst (IMS) using iron nitrate, manganese nitrate, and sodium silicate as precursors for the catalytic ozonation of AA. Additionally, we evaluated the stability and reusability of the IMS, as well as the mineralization of the AA after catalytic ozonation. We also propose potential degradation pathways of the IMS catalytic ozonation of AA.

## 2. Results and Discussion

### 2.1. Characterization of Synthesized IMS

SEM and EDS were used to determine the morphology and composition of IMS. As shown in Figure 1a,b, the surface of the IMS obtained for the ternary crystals was rough and comprised aggregates of nanorods. The EDS result (Figure 1c) shows that the main elements in the IMS were Fe, Mn, Si, and O. The Fe and Mn had a nearly 1.3:1 weight ratio, which did not conform to the theoretical element weight ratio of 1:1 in the material preparation. This was because the EDS results were only the black cross area in the SEM images (Figure 1a), where it can be seen that there were abundant dot-like crystallites embedded in the amorphous matrix. From this result, we can preliminarily infer that much more manganese oxide was embedded in the amorphous matrix than iron oxide. Figure 1d shows the XRD pattern of IMS. The highly dispersed silica sol generated by hydrolyzing sodium silicate inhibited the detection of the IMS peaks. The broad peak from 20° to 40° was typical amorphous silica [20].

Figure 2a shows the N_2_ adsorption–desorption isotherms of the IMS. It was determined that the IMS curves primarily corresponded to the type IV isotherm, with a hysteresis loop at relative pressures (P/P_0_) between 0.45 and 0.99. By using the BJH method, the pore diameter distribution of IMS was calculated (inset Figure 2a). Most pores in the IMS were less than 150 nm in diameter, but the pore size distribution was broad (2–1020 nm). The average pore diameter of the IMS was 30.79 nm, with a BET surface area of 437 m^2^/g and a pore volume of 0.49 cm^3^/g. Qi et al. [21] reported that a higher surface area and larger pore volume might improve the catalytic ozone performance by providing more active sites and facilitating the mass transfer of pollutants and ozone during catalytic ozonation.

The IMS was also analyzed by FTIR spectrum throughout the range 400–4000 cm^−1^ (Figure 2b). A broad and intense peak at 3452 cm^−1^ was attributed to the crystal structure’s O-H stretching vibration. The weak peak at 1690 cm^−1^ may be attributed to the stretching vibration of the -OH group from the physical adsorption of water molecules. The peaks at 1456 cm^−1^ and 1297 cm^−1^ were attributed to OH deformation vibrations of iron oxide and manganese oxide, respectively [17]. The band at 790 cm^−1^ was attributed to the Si-O bond. Meanwhile, the surface hydroxyl groups of IMS had a density of 27.8 mmol/L.

XPS analysis was performed on the IMS to determine the valence states of the Mn elements and Fe elements. As shown in Figure 3a, the IMS was composed of Fe, Mn, Si, and O elements. According to the Figure 3b spectra, the Mn 2p spectra of IMS presented two peaks of Mn 2p3/2 and Mn 2p1/2, respectively. The Mn 2p3/2 peak was divided into five peaks corresponding to Mn(III) (641.6 eV), Mn(IV) (642.9 eV), and the shake-up satellite peaks at 646.4 eV, 640.5 eV, and 644.2 eV suggested the presence of Fe-O-Mn, Mn-OH, and Mn-Si binary oxide [22]. Mn(IV) was present in the catalyst, allowing it to have high catalytic activity [23]. The O1s peaks (Figure 3c) with Be values of 532.3 eV, 531.3 eV, 530.5 eV, and 533.4 eV corresponded to H_2_O, Fe-O, Mn-O, and Si-O, respectively [24]. There were two components to the Fe 2p spectrum of the IMS: Fe 2p3/2 and Fe 2p1/2 (Figure 3d). There were three peaks associated with Fe 2p3/2. The first peak, 711.4 eV, was characteristic of Fe(III), and the second peak, 709.7 eV, was characteristic of Fe(II). The third peak at 713.2 eV was Fe-Si binary oxide [25]. The XPS results suggest that MnO_x_, FeO_x_, Fe-Si-Mn, Fe-Si, and Mn-Si binary oxide doped on IMS were multivalent. A study by Xing et al. [26] found that extensive electron transfer between different oxygenation states may accelerate the decomposition of ozone and induce the formation of highly reactive species.

### 2.2. Degradation of AA in the Presence of IMS

The removal of AA in different catalytic ozonation processes was evaluated as shown in Figure 4a. The addition of iron silicate (IS), manganese silicate (MS), and IMS increased the removal efficiency of AA to different degrees. Within 15 min, the IMS achieved the highest catalytic activity, removing AA at a rate of 98.9%. When IS or MS were present, the AA removal efficiency was 95.6% or 94.8%, respectively, whereas ozonation alone was only 62.7% efficient. As shown in Figure 4b, four different processes were also investigated to determine their effectiveness in removing the TOC. After a 15 min reaction, ozonation alone removed 31.9% of TOC compared with 58.9% by catalytic ozonation with IS and 52.1% by catalytic ozonation with MS. The enhancement of TOC removal was more pronounced in the presence of IMS as 70.3% of TOC was removed after 15 min. The results of the adsorption experiment (Figure 4a) show that after 15 min, AA adsorption onto IS, MS, and IMS was only 3.7%, 3.6%, and 4.4%, respectively. These results indicate that using different metal ions mixed with the catalyst preparation, improved the surface interface and electron transport capacity, thereby improving the catalytic performance. As opposed to adsorption by the catalysts, catalytic ozonation was the major cause of the increase in AA removal efficiency.

### 2.3. Stability of the IMS

To evaluate the recyclability and stability of IMS, five-cycle successive tests were carried out to test the removal of AA, and the results are shown in Figure 5. Because of the occlusion of the active sites by intermediates, there may be a minor inactivation of the catalyst. As the result of five consecutive catalytic cycles, the removal efficiency of AA decreased to 93.7% from 98.9%. Moreover, the treatment water contained no Fe ion and Mn ion above the detection limits, although the catalyst used for the five cycles still demonstrated better activity than the ozonation alone (AA removal efficiency of 62.7%). As a result, the IMS demonstrated fine stability and high catalytic activity in catalytic ozonation applications.

### 2.4. Analysis of the Mechanism

#### 2.4.1. Identification of the Dominant Oxidant

AA was negligibly absorbable by IMS, so, it was assumed that ozone and the catalyst were responsible for AA degradation and mineralization. A variety of processes were used to measure the efficiency of ozone utilization in solution to determine the effect of the catalyst on dissolved ozone concentrations (Figure 6a). It was estimated that the ozone utilization efficiency was calculated as AA removal multiplied by the adsorption removal per concentration of ozone utilized [16]. As shown in Figure 6a, compared with the ozonation alone, when catalysts were introduced the ratio considerably increased. After a 15 min reaction, the lowest ratio was 0.104 in the ozonation alone system. For the IMS catalytic ozonation process, it was 1.1 times and 1.16 times higher than the ratio for the IS and MS catalytic ozonation processes, respectively. Thus, the addition of IMS, IS, or MS accelerated the ozone decomposition, and IMS was more effective than IS and slightly better than MS. Based on these results, it was concluded that the surface hydroxyl groups of IMS as well as the Lewis acid sites could promote ozone degradation into ROS.

In order to verify the role of ROS in the catalytic ozonation of AA, TBA was used as an •OH scavenger. According to the results of Figure 6b, in the process of O_3_/IMS and ozonation alone, the presence of very low concentrations of TBA (0.1 mmol/L) had a negative effect on the degradation efficiency of AA in aqueous solution. The degradation efficiency of AA decreased to 47.3% as the concentration of TBA increased to 0.3 mmol/L, which was the same as the ozonation alone system with TBA. However, the AA degradation efficiency scarcely increased when the TBA concentration increased in both systems. As described in our previous report [27], this phenomenon is consistent with free radical properties. These results indicate that the catalytic removal of AA in the presence of IMS was mainly due to two possible paths: (1) the generation of •OH formed by the decomposition of ozone; or (2) direct reaction by ozonation alone. The electron paramagnetic resonance (EPR) technique was used to investigate the generation of •OH in the ozonation alone and catalytic ozonation processes. The ESR spectra obtained after ozonation alone and after catalytic ozonation are shown in Figure 6d. The catalytic ozonation process produced much stronger signals than ozonation alone. Hence, the presence of IMS in the ozonation system accelerated the generation of •OH in both the processes mentioned above.

According to Equations (1)–(4), the reaction of O_3_/•OH with unsaturated organics or two •OH combinations could produce H_2_O_2_ [28]. To confirm the H_2_O_2_ generated in the ozonation process, ozonation alone and IMS catalytic ozonation processes were investigated for H_2_O_2_ accumulation in an aqueous solution. As shown in Figure 6c, due to the decomposition of ozone H_2_O_2_ was formed in the two systems. In both processes, the concentration of H_2_O_2_ increased with the reaction time and reached a maximum before decreasing. Moreover, the concentrations of H_2_O_2_ were greater in the IMS catalytic ozonation process than in ozonation alone. This indicated that the addition of IMS promoted the generation of H_2_O_2_. The AA mineralization efficiency was improved by the H_2_O_2_ decomposition of O_3_ to generate •OH.
O_3_ + HO^−^ = HO_2_^−^ + O_2_,(1)
HO_2_^−^ + H = H_2_O_2_,(2)
2O_3_ + H_2_O_2_ = 2•OH + 3O_2_,(3)
•OH + •OH = H_2_O_2_(4)

#### 2.4.2. Effect of Ions and Water Quality Background on the Removal of AA

As a strong Lewis base, PO_4_^3−^ competed with the ozone for the surface active site on the catalysis surface and substituted the hydroxyl radical inhibiting decomposition of the ozone [24]. The Cl^−^ and HCO_3_^−^ molecules possess a high affinity for surface active sites, and can quickly occupy the catalyst surface, resulting in a reduction in the catalytic efficiency and ozone decomposition. The alkali metals Ca^2+^ and Mg^2^+ have stable valences and do not participate in degradation reactions [29]. AA removal was observed when PO_4_^3−^, Cl^−^, HCO_3_^−^, Ca^2+^, and Mg^2+^ ions were added to the reaction system (Figure 7a–e). As shown in Figure 7, the presence of PO_4_^3−^ inhibited the catalytic ozonation of AA. The AA removal rate constantly decreased from 98.7% to 50.3% as the PO_4_^3−^ concentration increased from 0 to 0.5 mmol/L, suggesting surface hydroxyl groups and Lewis acid sites on IMS were the active sites in the catalytic ozonation process, and •OH plays a crucial role in AA degradation. With the increase in the Cl^−^, HCO_3_^−^, Ca^2+^, and Mg^2+^ concentrations from 0 to 0.5 mmol/L, the AA degradation efficiency decreased to 56.7%, 61.2%, 82.3%, and 77.0%, respectively. The presence of Cl^−^ and HCO_3_^−^ in aqueous solution might consume the •OH quickly, so, the degradation efficiency of AA decreased [24]. The •OH may be captured by the coexistent chloride ions, which may explain the decreased AA in the presence of Ca^2+^ and Mg^2+^. These results directly illustrate that the surface hydroxyl groups, •OH in aqueous solution, and Lewis acid sites on IMS contribute to catalytic ozonation.

Further investigation of the water quality background effects on AA degradation was conducted using river water with the following parameters: TOC: 2.9 mg/L, pH 7.0, UV_254_ 0.033, and NTU 0.5. As shown in Figure 7f, after 15 min the removal efficiency of AA in the river water was 73.3%, and that in tap water was 78.2%. The reason for this is that natural water has rigidity, alkalinity, and an abundance of matrixes, such as those formed by natural organic matters (NOMs). NOM and alkalinity could react with •OH and become the main inhibiting factors for •OH in aqueous solution [16].

#### 2.4.3. Catalytic Mechanism Analysis

In the heterogeneous catalytic ozonation process, catalytic ozonation has three main mechanisms: (1) direct interaction of dissolved ozone molecules; (2) radical degradation in bulk reactions; and (3) O_3_ molecules interacting with the catalyst and generating radicals indirectly [30]. In above section, we showed that IMS was an effective catalyst for the catalytic ozonation of AA. According to the results of the XRD, XPS, and FTIR (Figure 1, Figure 2 and Figure 3), we know that the IMS was mainly composed of MnO_x_, FeO_x_, Fe-Si-Mn, Fe-Si, and Mn-Si binary oxide, which has a large specific surface area and surface hydroxyl content. He et al. reported that the surface hydroxyl groups in the heterogeneous catalytic ozonation process could provide a surface-active field and accelerate the chain decomposition of the ozone to produce •OH [22]. Numerous studies have shown that MnO_x_ [22], FeO_x_ [30], Fe-Si [9], and Mn-Si binary oxide [17] can catalyze ozone decomposition to produce •OH. The ESR spectra (Figure 6d) showed that •OH was generated both in the ozonation alone and catalytic ozonation, and the IMS enhanced ozone decomposition to generate more •OH than ozonation alone. The TBA experiment results indicated that •OH played an important role in the ozonation process, and AA could be degraded by •OH and O_3_. As determined by the ion test, the Lewis acid sites and surface hydroxyl groups on IMS were the active sites during the catalytic ozonation process.

According to the experimental results, the reaction mechanism of the IMS catalytic ozonation is proposed. First, the AA molecules can be directly oxidized by O_3_ in the solution. Second, hydroxyl groups and Lewis acid sites on the surface of the IMS can accelerate •OH generation from ozone decomposition, consequently enhancing the degradation of AA.

## 3. Materials and Methods

### 3.1. Materials

AA was purchased from Tianjin Chemical Factory (Tianjin, China) with a purity of >99.5%. Methyl alcohol and borax buffer used for high-performance liquid chromatography were HPLC grade (Sigma Aldrich, St. Louis, MO, USA). 5, 5-Dimethyl-1-pyrroline N-oxide (DMPO, >97%) was purchased from USA Chem service (St. Louis, MO, USA). Other chemicals used included ferric nitrate, manganese nitrate, sodium silicate, sodium thiosulfate, hydrochloric acid, sodium hydroxide, tert-butanol (TBA), sodium chloride, calcium chloride, sodium bicarbonate, phosphoric acid, and so on; the reagents used in the experiments were analytical grade and did not require further purification. The entire experimental process was carried out with distilled water.

### 3.2. Synthesis of Catalysts

The IMS catalyst was synthesized using Mn(NO_3_)_2_, Fe(NO_3_)_3_, and Na_2_SiO_3_ as the precursor materials. First, 179 g Mn(NO_3_)_2_ and 242 g Fe(NO_3_)_3_ were dissolved in 1 L of distilled water. A solution of Na_2_SiO_3_ was then added into the solution under magnetic stirring. The dropping of the Na_2_SiO_3_ solution ceased when the pH of the suspension reached 7.0. By adding 0.01 mol/L NaOH into the solution, the pH of the suspension was adjusted to 12, and the solution was incubated for 24 h at 40 °C. After collecting the precipitate, ultra-pure water was used to wash the precipitate several times until the pH and conductivity of the supernatant were maintained. The final step in the process was to dry and grind the precipitate at 60 °C. It was 100% reproducible from run to run for the as-synthesized IMS catalyst.

In accordance with our previous publications, iron silicate (IS) and manganese silicate (MS) were prepared [9,17].

### 3.3. Ozonation Procedure

AA degradation experiments were conducted at ambient temperature 20 °C in a semicontinuous flow mode. A 1.2-L Florence flask reactor was used as the reactor. The reactor was filled with 1000 mL water for each experiment. A CF-G-3-010 g ozone generator (Guolin, Qingdao, China) was used to produce ozone, which was generated using pure oxygen as the gas source. In a typical catalytic degradation procedure a certain amount of IMS catalyst and AA were mixed in a flask under stirring control. Then, ozone gas was continuously fed into the bottom of the flask through a diffuser made of silica bubbles at the bottom of the reactor while it was constantly stirred. After each interval (0, 1, 3, 5, 10, and 15 min), 5.0 mL samples were taken from the reactor, quenched by adding Na_2_S_2_O_3_ solution (0.01 mol/L), and centrifuged to analyze the residual concentration of AA. IMS adsorption and ozonation alone (without catalyst) experiments were conducted under the same conditions.

A batch experiment was conducted to test the ozone utilization efficiency. In the experiment, 1000 mL ultra-pure water was filled into the reactor. As a control, the initial concentration of aqueous ozone was controlled at 1.0 mg/L, the catalyst dosage was 500 mg, and the initial concentration of AA was also 500 mg/L. To begin the experiment, the magnetic stirrer was turned on, and the reactor was sealed. The residual dissolved ozone was measured by sampling 5 mL of the solution after a designated interval. Each experiment was repeated three times and the averages and error bars are shown in the figures.

### 3.4. Analytical Methods

LC-1200 high-performance liquid chromatography (HPLC) (Shimadzu, Tokyo, Japan) with 4.6 mm × 250 mm C18 water columns was used at room temperature to determine the concentration of AA. Then, 0.5 mL/min of methyl alcohol and borax buffer (7:3, v:v) were pumped simultaneously to elute the sample of AA. A volume of 1.0 mL was used for the injection. A UV–vis detector (Shimadzu, Tokyo, Japan) with wavelengths set at 230 nm was used. The concentration of ozone in the gas was measured using the indigo method. A TOC-VCPH analyzer (Shimadzu, Tokyo, Japan) was used to analyze the total organic carbon (TOC) based on carbon dioxide infrared absorption. X-ray diffraction (XRD) was carried out on a Bruker D8 Advance Diffractometer with Cu Kα radiation (λ = 1.5418 Å). Energy dispersive spectrometry (EDS) was conducted on Genesis (Input Cokv Zokw Co.Ltd., Tokyo, Japan). Scanning electron microscopy (SEM, Quanta 200FEG, FEI Corporation, Co.Ltd., Tokyo, Japan) was used to analyze the surface morphology of the catalyst. X-ray photoelectron spectroscopy (XPS, PHI 5700, PerkinElmer, Waltham, MA, USA) was used to analyze the surface chemical composition of the catalyst. A PerkinElmer Paragon1000 FTIR spectrometer with a spectral range of 4000–400 cm^−1^ was used to measure the Fourier transform infrared spectroscopy (FT-IR, Spectrum One, Shimadzu, Tokyo, Japan). A Surface Area and Porosity Analyzer (Micromeritics ASAP 2020, Atlanta, GA, USA) was used to measure the BET surface area. An inductively coupled plasma atomic emission spectrometer (Optima 5300 DV, Perkin Elmer, Waltham, MA, USA) was used to measure the metals leached. The saturated deprotonation method was used to analyze the density of the surface hydroxyl groups [31].

## 4. Conclusions

IMS was synthesized using Mn(NO_3_)_2_, Fe(NO_3_)_3_, and Na_2_SiO_3_ as the precursor materials. XRD, FTIR, XPS, and BET analyses confirmed that the Fe-O-Mn bond, Fe-Si, Mn-Si binary oxide, and an abundance of functional groups were formed on the surface of the IMS. In the catalytic ozonation processes, IS, MS, and IMS were all more effective than ozone alone in the degradation of AA, and IMS was slightly better. The IMS showed the highest mineralization rates of AA. During the IMS catalytic ozonation, the IMS adsorption had no effect on TOC removal or AA degradation. It has been found that IMS can activate the decomposition of ozone and generate H_2_O_2_ and •OH. The inhibition of the hydroxyl radical scavenger on the catalytic ozonation indicated that it was a combination of the hydroxyl radical reaction and the ozonation process that removed AA. After five cycles of testing, the IMS catalyst remained stable and active.

## Figures and Tables

**Figure 1 molecules-27-04973-f001:**
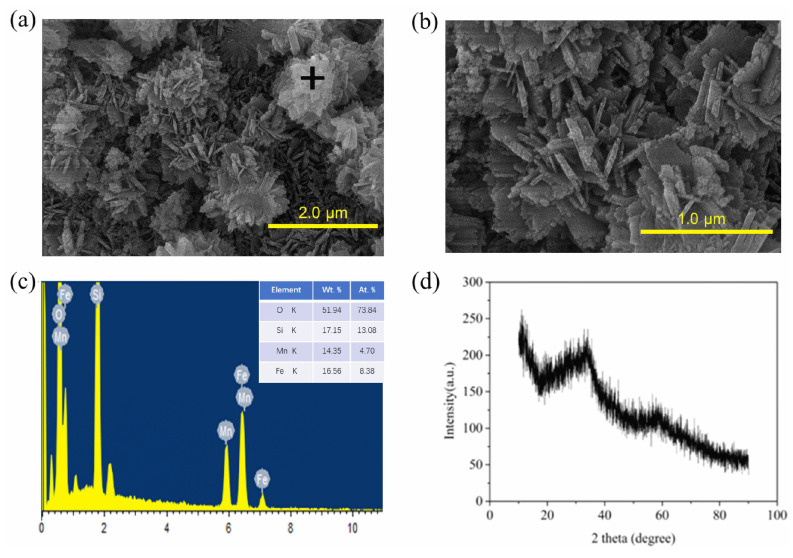
SEM micrographs (**a**,**b**), (**c**) EDS spectra of the black cross in (**a**), and XRD spectra of IMS (**d**).

**Figure 2 molecules-27-04973-f002:**
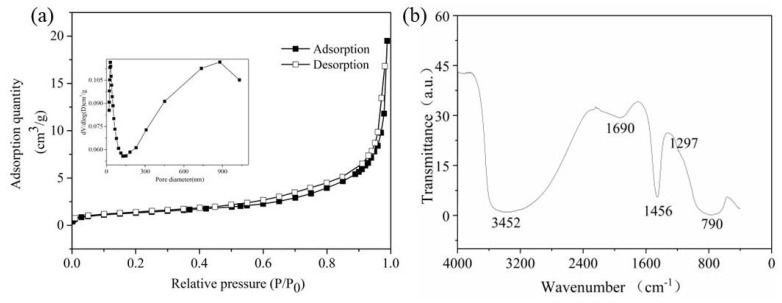
Nitrogen adsorption–desorption isotherm (the inset shows its BJH pore size distribution curve) (**a**) and FTIR spectrum (**b**) of the IMS.

**Figure 3 molecules-27-04973-f003:**
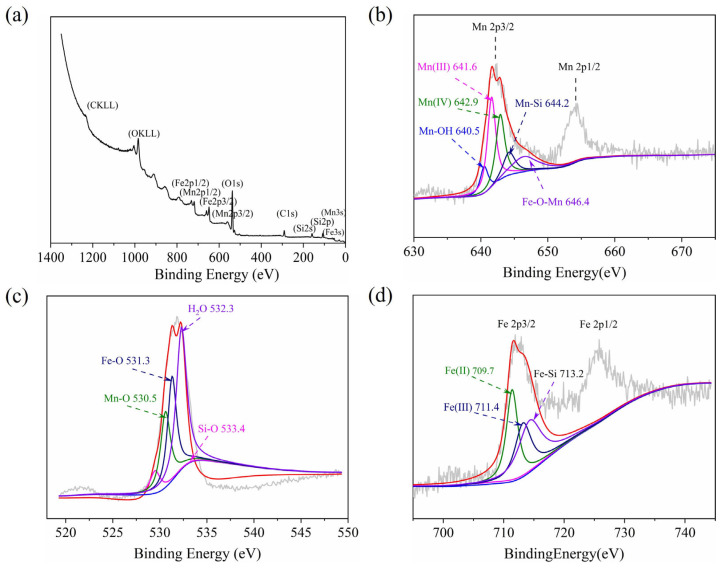
XPS spectra of the IMS composites (full scan) (**a**), Mn 2p spectra (**b**), O 2s spectra (**c**), and Fe 2p spectra (**d**).

**Figure 4 molecules-27-04973-f004:**
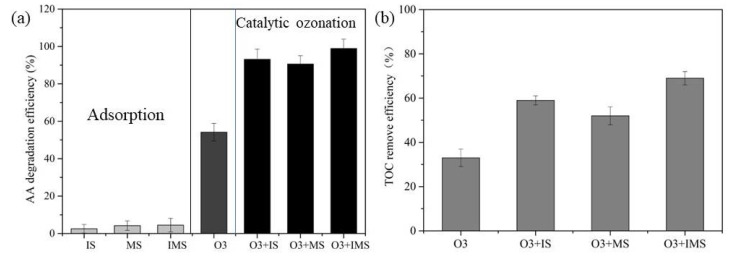
Comparation of AA removal in different processes (**a**); TOC removal efficiency (**b**). Experiment conditions: [AA]_0_ = 1000 mg/L, [O_3_]_0_ = 15 mg/min, [catalyst dose]_0_ = 500 mg/L, flow rate of oxygen 0.6 L/min, reaction temperature 20 °C.

**Figure 5 molecules-27-04973-f005:**
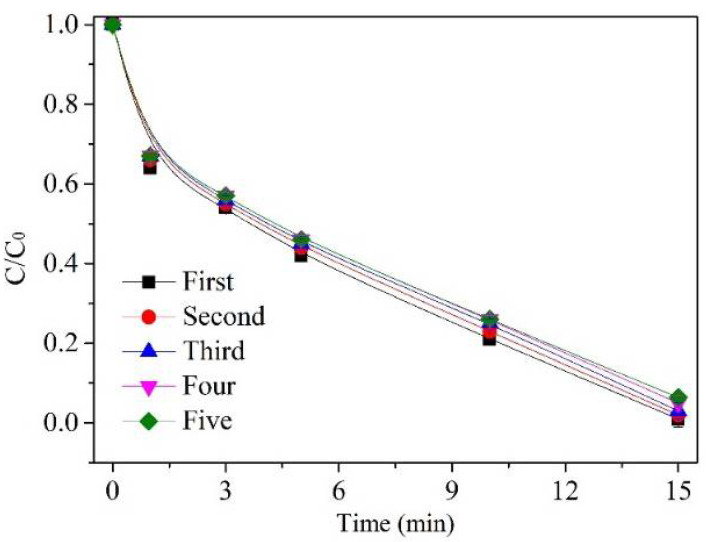
AA degradation efficiency. Experiment conditions: [AA]_0_ = 1000 mg/L, [O_3_]_0_ = 15 mg/min, [catalyst dose]_0_ = 500 mg/L, flow rate of oxygen 0.6 L/min, reaction temperature 20 °C.

**Figure 6 molecules-27-04973-f006:**
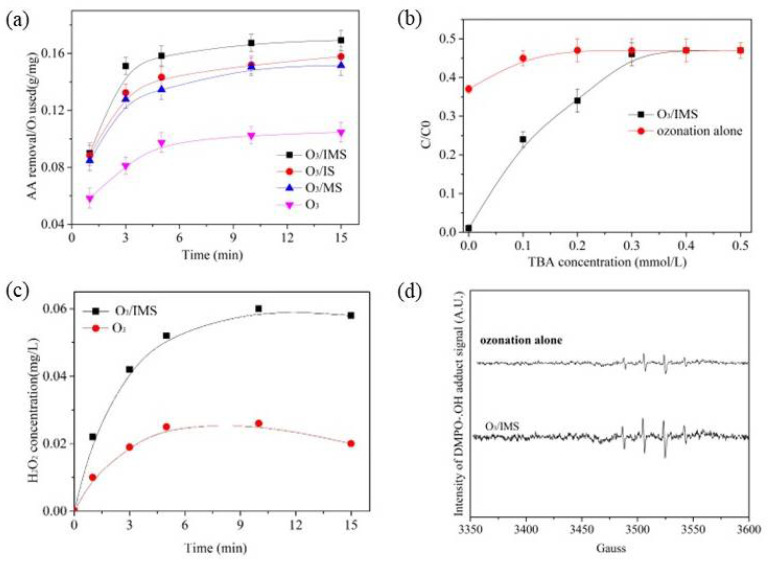
Effective ozone use ratio in the ozonation process (**a**), effect of TBA on the degradation of AA (**b**), the concentration of H_2_O_2_ in different process (**c**), spectra of DMPO-OH signals in different processes (**d**). Experiment conditions: [AA]_0_ = 500 mg/L, [O_3_]_0_ = 1.0 mg/L, [catalyst dose]_0_ = 500 mg/L, [DMPO]_0_ = 100 mmol/L.

**Figure 7 molecules-27-04973-f007:**
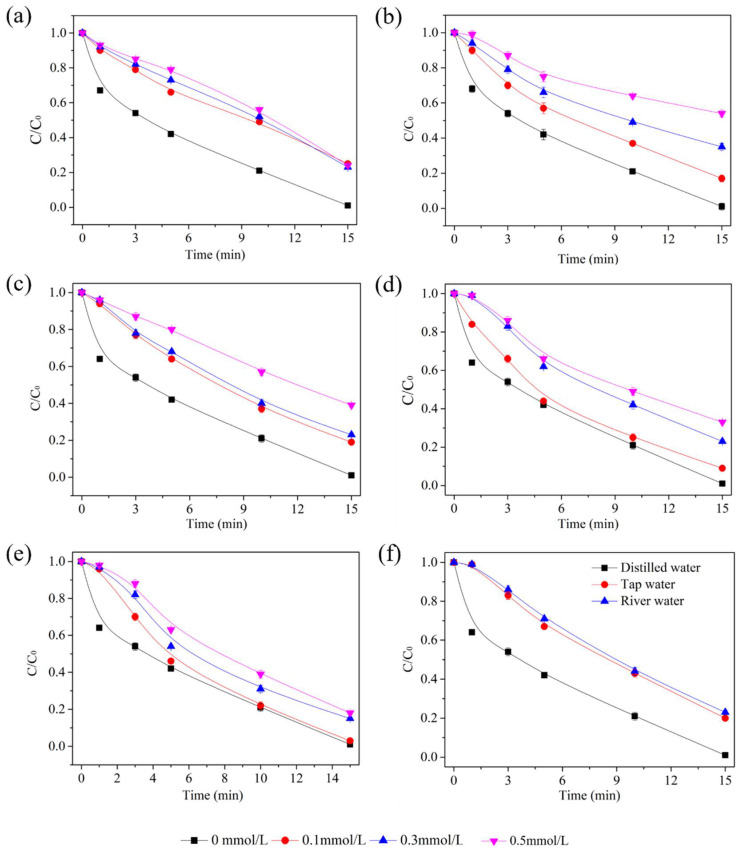
Effect of inorganic anions including PO_4_^3−^ (**a**), Cl^−^ (**b**), HCO_3_^−^ (**c**), Ca^2+^ (**d**), Mg^2+^ (**e**), and the effect of water quality background (**f**). Experiment conditions: [AA]_0_ = 1000 mg/L, [O_3_]_0_ = 15 mg/min, [catalyst dose]_0_ = 500 mg, flow rate of oxygen 0.6 L/min, reaction temperature 20 °C.

## Data Availability

The data presented in this study are available on request from the corresponding author. The data are not publicly available due to privacy.

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
