# Peer review of "Heterogeneous Catalysis of Ozone Using Iron–Manganese Silicate for Degradation of Acrylic Acid"

_molecules, 2022, doi:10.3390/molecules27154973_

Round 1

Reviewer 1 Report

I have read the manuscript “Heterogeneous catalysis of ozone using iron-manganese silicate 2 for degradation of acrylic acid” carefully. The manuscript is well organized but there still some points that can be improved. Follows some specific comments.

The English is very poor.

Line 16. Rewrite the sentence “The removal efficiency of AA was enhanced highest in IMS catalytic ozonation processes 16 (98.9%) in 15min, compared with ozonation alone (62.7%) and iron silicate (IS) catalytic ozonation 17 (95.6%) and manganese silicate catalytic ozonation (94.8%)” to be “The removal efficiency of AA was highest in IMS catalytic ozonation processes (98.9% in 15 min), compared with ozonation alone (62.7%), iron silicate (IS) catalytic ozonation (95.6%) and manganese silicate catalytic ozonation (94.8%)”.

Keywords should be different from the title.

Please add a reference to the sentence “As a result, they are frequently detected in surface water and wastewater”

The authors state “It was found that Fe and Mn had nearly a 1:1 atomic ratio” while the atomic ratio shown in figure 1c are 4.7 and 8.38 for Mn and Fe, respectively. Please clarify.

Please refer to figure 3c in the text.

In the caption of figure 4 and 5 give the dosage of catalyst in mg/L not mg only.

Line 122. The authors state “After 15 min reaction, the TOC removal rate was 70.3% in the IMS catalytic ozonation process which was somewhat more efficient than the other three”. Please give the value of TOC removal for other treatment processes. Also, please note that the differences in the removal of AA by IS, MS and IMS were insignificant. Therefore, either IS or MS will be better than IMS from the economic point of view. Please explain and give more discussion on this point.

Page 5. The authors state “it was 1.16 times and 1.1 times higher than the ratio for IS and MS catalytic ozonation processes, respectively”. According to the results displayed in figure 6a, these ratios seem reversed, it should be “1.16 times and 1.1 times higher than the ratio for MS and IS catalytic ozonation processes, respectively”

Page 5. The authors state “and IMS could do so more effectively”. Please change to “and IMS could do more effectively than IS and slightly better than MS”

Page 6. Define ESR at the first time.

The legend in figure 6 is incorrect.

Line 171. It should be “river water”.

Line 173. the authors state “that natural water is rigid” what the authors meant by “rigid”.

Support section 2.4.3. with some relevant citations.

The conclusion section should show that IS, MS and IMS are all more effective than ozone alone and that IMS is slightly better.

In the synthesis of catalyst section give exact weights of Mn(NO3)2 and Fe(NO3)3.

Author Response

Dear  reviewers,

On behalf of my co-authors, we thank you very much for giving us an opportunity to revise our manuscript. We appreciate the editor and reviewers very much for their patience, constructive comments and suggestions on our manuscript. Thank you very much for giving us the opportunity to revise and resubmit our manuscript. The comments and suggestions from the reviewers are encouraging and insightful, which are of great value to improve the quality of our work. We have addressed all these major points and other issues carefully and revised the manuscript accordingly. We provide detailed and point-by-point responses to the reviewers’ comments in the following pages. Note that the reviewers’ comments are presented in black font, and our responses are in blue font, and all changes made in the manuscript are marked in red font. Please let me know if you have any further questions. We hope the revision will be satisfactory.

Sincerely,

Yue Liu

School of Energy & Environment Engineering, Zhongyuan University of Technology, Zhengzhou, 451191, P R China

Reviewer 2 Report

This manuscript as well as the previously published papers (Liu et al. Appl. Catal. A 403(1-2) (2011) 112; Liu et al. Sep. Purif. Technol. 172 (2017) 251; Liu et al. Catal. Commun. 89 (2017) 81; Liu et al. RSC Adv. 10(15) (2020) 9146 and etc.) reports the data on catalytic performance of metal-silicates (Zn-Cu, Zn-Fe, Fe-Mn and etc.) in degradation of organic pollutants by ozone in water. It was shown that iron-manganese silicate catalyst demonstrates high catalytic activity and stability in ozonation of acrylic acid.

The work is well built. The experimental data are clearly presented and commented. The conclusions are consistent with the results obtained. The topic is perfectly in line with the “Catalysts” journal. I recommend it for publishing after the following correction to be made:

1) The English in the present manuscript is required major improvement.

Author Response

(The authors gave the same response as above.)

Round 2

Reviewer 1 Report

I have read the revised version of the manuscript “Heterogeneous catalysis of ozone using iron-manganese silicate 2 for degradation of acrylic acid” carefully. The authors response to my comment is satisfactory, the manuscript has been improved over the original and the manuscript can be accepted in the current form.